# Micellization of Polystyrene-*b*-Polyglycidol in Dioxane and Water/Dioxane Solutions

**DOI:** 10.3390/polym12010200

**Published:** 2020-01-13

**Authors:** Lukasz Otulakowski, Andrzej Dworak, Aleksander Forys, Mariusz Gadzinowski, Stanislaw Slomkowski, Teresa Basinska, Barbara Trzebicka

**Affiliations:** 1Centre of Polymer and Carbon Materials, Polish Academy of Sciences, M. Curie-Sklodowskiej 34, 41-819 Zabrze, Poland; lotulakowski@cmpw-pan.edu.pl (L.O.); adworak@cmpw-pan.edu.pl (A.D.); aforys@cmpw-pan.edu.pl (A.F.); 2Centre of Molecular and Macromolecular Studies, Polish Academy of Sciences, Sienkiewicza 112, 90-363 Lodz, Poland; mariuszg@cbmm.lodz.pl (M.G.); staslomk@cbmm.lodz.pl (S.S.); basinska@cbmm.lodz.pl (T.B.)

**Keywords:** polystyrene-*b*-polyglycidol, amphiphilic block copolymer, micellization, aggregation, critical micelle concentration, critical water content

## Abstract

In this work, the self-assembly of a series of amphiphilic polystyrene-*b*-polyglycidol (PS-*b*-PGL) diblock copolymers in dioxane and dioxane/water mixtures is presented. The PS-*b*-PGL have an average degree of polymerization (DP) of PS block equal to 29 units and varied degrees of polymerization for the glycidol segments with DPs of 13, 42, 69 and 117. In dioxane, amphiphilic diblock copolymers form micelles with the hydrophilic PGL placed in the core. Critical micelle concentration (CMC) was determined based on the intensity of scattered light vs. concentration. The micelle size was measured by dynamic light scattering and transmission electron microscopy. Also, the behaviour of the copolymer was studied in water/dioxane solutions by following the changes of scattered light intensity with the addition of water to the system. Critical water content (CWC) of the studied systems decreased as the initial PS-*b*-PGL concentration in dioxane increased. This process was accompanied by a decrease in the size of aggregate formed. For a given initial copolymer concentration, the size of copolymer aggregates decreased linearly with increasing the length of the PGL block

## 1. Introduction

In recent years, the solution behaviour and applications of amphiphilic block copolymers have been intensively studied. Amphiphilic block copolymers spontaneously self-assemble in water and in non-polar solvents, which are good solvents for hydrophilic and hydrophobic blocks, respectively. Frequently, copolymer aggregates are spherical. Core-corona micelles formed in water or, in a broader sense, in good solvents for the hydrophilic blocks, are called regular. Micelles formed in non-solvents for the hydrophilic block are called “reverse” micelles with cores composed of hydrophilic chains. Aggregation of amphiphilic block copolymers is governed by their chemical structure and the length of blocks as well as by solution parameters. The formation of diverse morphologies with high ordering of copolymer chains such as cylinders, vesicles, wormlike structures, bilayers, and others have been reported [1,2,3,4]. The process of supramolecular object formation is explained by taking into account micelle free energy, which consists of free energy of the core, the core-corona interface, and the corona-continuous phase [5]. The use of block copolymer colloidal systems has been reported in various fields like controlled drug delivery [6,7], carriers of biological markers [8], separation processes [4], and many others.

To induce self-assembly, a selective solvent is often added to the solution of amphiphilic copolymers dissolved in a solvent common for both blocks. The last procedure was systematically elaborated upon by Eisenberg’s group to induce aggregation of polystyrene-*b*-poly(acrylic acid) dissolved in THF, DMF, or dioxane by adding water [5,9,10,11]. They measured the critical water content (CWC) at which phase separation starts, and they related it to copolymer molar mass, copolymer concentration in solution, and the nature of the initial solvent. Polystyrene-*b*-poly(acrylic acid) yielded an aggregated system of different morphologies that was related to changes in the stretching of the polystyrene block in the core region. The process can be controlled by varying the initial copolymer concentration and the solvent composition. 

Research on polystyrene-*b*-poly(ethylene oxide) (PS-*b*-PEO) micellization began many years ago. Self-assembly of PS-*b*-PEO leads mainly to spherical core-shell micelles with a PS core and a PEO shell [12,13,14,15,16,17]; sometimes it is accompanied by formation of larger aggregates [16,17]. Vesicles and worm-like structures formed by PS-*b*-PEO in water have also been reported [18]. 

Aggregation of PS-*b*-PEO was studied in mixtures of solvents with one being the selective solvent for PEO. Eisenberg et al. [19] observed various morphologies created by PS-*b*-PEO in water/DMF solution. Aggregation and morphological transitions of PS-*b*-PEO were investigated in DMF/water and in DMF/acetonitrile by Cheng’s group [20,21,22]. Micelle morphologies changed from stars to worm-like and finally to vesicles when a solvent selective for the PEO block was added to the DMF copolymer solution. The authors also observed morphological changes of PS-*b*-PEO in DMF/water that were induced solely by temperature [20].

Reverse spherical micelles containing PEO blocks as the micellar core and PS blocks as the corona were formed in a dioxane/cyclohexane mixture at room temperature [23]. PS_164_-*b*-PEO_827_ formed aggregates of about 30 nm as measured by TEM. The studies showed that with decreasing temperature, the micelles transformed into a mixture of spheres and cylinders. At lower temperatures, only cylinders were formed, and at −5 °C a mixture of cylinders and vesicles was observed. 

Poly(ethylene oxide) chains can be replaced in an amphiphilic block copolymer with poly(2,3-epoxy-1-propanol) (polyglycidol) in which each repeating unit contains an additional methylhydroxyl group. The self-consistent reaction field calculations revealed that the additional groups in the chain repeating units cause a decrease in the free enthalpy of hydration of the most stable trimers of polyglycidol compared to the free enthalpy of hydration of PEO trimers ended with hydroxyl groups [24]. This means that the polyglycidol macromolecule is much more prone to hydration than PEO. Pure polyglycidol is easily soluble in solvents with high polarity like DMF or water; whereas in most organic solvents, like benzene, THF or dioxane, its solubility is highly limited.

A few years ago, the synthesis of amphiphilic polystyrene-*b*-polyglycidol (PS-*b*-PGL) copolymers was elaborated [25,26]. Protection and deprotection of the hydroxyl group in glycidol allowed for the synthesis of various macromolecular architectures (e.g., linear or hyperbranched structures containing polyglycidol) [27,28,29,30,31]. Structures based on polyglycidol can serve as nanocontainers [32,33,34,35], nanolayers for cell harvesting [36], or layers resistance against protein adsorption [24,37]. Polyglycidol seems to be an interesting alternative to PEO in amphiphilic polymer structures.

The micellization of linear block copolymers containing a polyglycidol block has been studied. Dimitrov et al. [38] followed the micellization of polyglycidol-*b*-poly(ethylene oxide)-*b*-poly(DL-lactide) (PGL-*b*-PEO-*b*-PLA) terpolymers with different block lengths to measure CMC and the size of the micelles. Toncheva-Moncheva et al. [39] synthesized a series of polyglycidol-*b*-poly(ε-caprolactone)-*b*-polyglycidol copolymers by “click” chemistry. Different protocols employed in micelle preparation influenced the sizes of the terpolymer aggregates in solution. Gadzinowski et al. [40] prepared a series of PEO-*b*-PGL-*b*-PLA triblock copolymers, which formed micelles with diameters ranging from 21 to 37 nm. PGL blocks bearing hydroxyl groups could undergo crosslinking and thus stabilize the micelles formed. Functionalization of the polyglycidol block in the above-mentioned copolymer by reaction with 4-(phenyl)azo benzoic acid yielded a photosensitive copolymer with its CMC depending on sample irradiation with UV light with wavelengths in the range of 300–400 nm [41].

The reverse spherical micelles of PS-*b*-PGL copolymers with different numbers of units in the blocks were studied in toluene by Siebert et al. [42]. It was shown that with decreasing the PGL/PS ratio, the CMC of the copolymers increased. The longer the PGL chain, the larger were the particle diameters. The micelles′ PGL core was crosslinked with titanium tetraisopropoxide. 

Micellization of PS-*b*-PGL copolymers in water was reported in our previous work [43]. Copolymer chains consist of a constant number of styrene units and various lengths of PGL blocks. When dialyzed from dioxane to water, the copolymers mainly formed regular spherical micelles with only a small population of clusters in case of the shortest PGL blocks. The values of the critical micelle concentrations increased linearly with the PGL/PS ratio.

Here we describe the behaviour of amphiphilic PS-*b*-PGL copolymers with a constant average degree of polymerization of polystyrene block and a precisely controlled average degree of polymerization of polyglycidol blocks in dioxane and dioxane/water solutions. The aim of our work was to determine the influence of water (good solvent for PGL and nonsolvent for PS) on the micellization process of PS-*b*-PGL. The micellization of copolymers in dioxane was considered as a reference. Our study was focused on the effect of variation of PGL length on critical micelle concentration in dioxane and on critical water content in dioxane/water mixtures. 

According to our knowledge, the self-assembly of amphiphilic PS-*b*-PGL copolymers has never been investigated in dioxane and in dioxane/water mixtures. 

## 2. Materials and Methods

### 2.1. Materials

#### 2.1.1. Reagents

Styrene (Sigma-Aldrich, Poznan, Poland) was purified by keeping it over calcium hydride, and it was distilled under a vacuum. The purified monomer was stored in a refrigerator. Toluene (Chempur, Piekary Slaskie, Poland) was purified and dried by stirring it with added sulfuric acid, neutralizing with a NaHCO_3_ water solution and it was dried over MgSO_4_ and by boiling under reflux over sodium. The required portions of toluene were distilled just before using. Ethylene oxide (Fluka) was dried by keeping it over calcium hydride, and it was distilled under a vacuum and stored at 4 °C before being used. Tetrahydrofuran (THF, Chempur, Piekary Slaskie, Poland) was purified from traces of peroxides by keeping it for a few days over potassium hydroxide. Thereafter, THF was distilled under a vacuum and stored over metallic sodium. Portions of THF were distilled, placed in ampoules containing a sodium-potassium alloy, and stored for future use. 1-Ethoxyethyl glycidyl ether (GLB) was synthesized as described in Refs. [44,45]; then it was distilled under reduced pressure and dried over calcium hydride. sec-Butyllithium (1.4 M solution in cyclohexane, Sigma-Aldrich), naphthalene 99% (Sigma-Aldrich), aluminium chloride hexahydrate, 1,4-dioxane, and metallic potassium (Sigma-Aldrich) were used as received. 1,4-dioxane (Sigma-Aldrich) was used as received. The water used to obtain the polymer solutions was purified using a commercial ion exchange system (Hydrolab, Straszyn, Poland).

#### 2.1.2. Synthesis of PS-*b*-PGL Copolymers

PS-*b*-PGL block copolymers, were synthesized as described in detail in our previous work [43].

Briefly, in the first step, polystyrene with a targeted degree of polymerization (29) was synthesized by anionic polymerization using *sec*-butyllithium as an initiator. The acquired hydroxyl-terminated polystyrene (PS-macroinitiator) was precipitated into methanol and dried under a vacuum.

PS-*b*-PGL was synthesized by deprotection of polystyrene-*b*-poly(1-ethoxyethyl glycidyl ether) copolymers (PS-*b*-PGLB). The PS-*b*-PGLB copolymer was prepared by polymerization of GLB on the PS-macroinitiator. The macroinitiator was first dissolved in dry THF and activated by potassium napthalenide solution. Then, depending on the desired molar mass of the PGLB block, various amounts of GLB were added. The resulting copolymers were precipitated into warm water. The crude viscous products were dried under vacuum, dissolved in 1,4-dioxane, and subsequently lyophilized. 

Deprotection of the 1-ethoxyethyl glycidyl ether units in the copolymers was performed according to [27]. PS-*b*-PGLB copolymer was dissolved in 1,4-dioxane and subsequently methanol was added dropwise until the mixture become slightly opaque. Then, AlCl_3_·6H_2_O was added and the mixture was stirred for 3 h. Finally, the copolymer was dialyzed against water. The resulting block copolymers (Scheme 1) are designated as PS-*b*-PGL(X), where X denotes the sample number from 1 to 4. 

### 2.2. Methods

#### 2.2.1. Formation of Micelles in Dioxane and CMC Measurements

To determinate the CMC of the PS-*b*-PGL(X) copolymers in dioxane, a series of solutions were prepared in dioxane by direct dissolution of the copolymers. The concentration of copolymers varied from 2 to 0.01 mg/mL. The solutions were shaken on a shaker for two days at room temperature and stabilized for one day before measurements. Then, the solutions were placed in DLS vials. The CMC was determined from the intensity of scattered light vs. solution concentration as the point of intersection of the tangents of the obtained curve. 

#### 2.2.2. Formation of Aggregates in Water/Dioxane and CWC Measurements

A series of copolymer solutions in dioxane with different concentrations in the range from 10^−3^ to 10^−5^ g/g was prepared. A quantity of 1.034 g of these solutions were placed in the DLS test tube, and then portions of water in an amount of 5% by weight were subsequently added. The dependence of the intensity of scattered light and solution concentration was analysed based on DLS measurements to determine the CWC. Measurements were carried out at room temperature.

#### 2.2.3. Characterization by Dynamic Light Scattering

DLS was used to measure the CMC of polymers in water solution, CWC of polymers in water/dioxane solution, and the sizes of the aggregated structures. DLS studies were performed using a Brookhaven BI-200 goniometer (Brookhaven Instruments, Holtsville, NY, USA) with vertically polarized incident light of wavelength *λ* = 632.8 nm (He-Ne laser, 35 mW) and equipped with a Brookhaven BI-9000 AT digital autocorrelator. The intensity of scattered light was measured for different concentrations at angle of 90°. The autocorrelation functions were analysed using the constrained regularized CONTIN method to obtain distributions of relaxation rates (*Γ*). The latter provided distributions of the apparent diffusion coefficient *D* (*D* = *Γ*/*q*^2^, where *q* is the magnitude of the scattering vector, *q* = (4*πn*/*λ*)sin(*θ*/2), and *n* is the refractive index of the medium). The apparent hydrodynamic radius ($R_{h}^{90}$) was obtained from the Stokes-Einstein equation (Equation (1))
(1)$$R_{h}^{90}=\frac{kT}{6\pi \eta D}$$
for *θ* = 90° where *k* is the Boltzmann constant and *η* is the viscosity of water at temperature *T*. The dispersity of particle diameters was given as $\mu_{2}/{\bar{\Gamma }}^{2}$, where $\bar{\Gamma }$ is the average relaxation rate and *µ*_2_ is its second moment.

#### 2.2.4. Transmission Electron Microscopy

Transmission electron microscopy images were obtained using a Tecnai F20 X TWIN microscope (FEI Company, Hillsboro, Oregon, USA) equipped with a field emission gun, operating at an acceleration voltage of 200 kV. Images were recorded on an Eagle 4k HS camera (FEI Company, Hillsboro, Oregon, USA) and processed with TIA software (FEI Company, Hillsboro, Oregon, USA).

A quantity of 6 μL of solution was placed on a copper grid covered with carbon film and air dried at room temperature before measurements.

## 3. Results and Discussion

Polystyrene with –CH_2_CH_2_O^−^K^+^ end-groups and DP = 29 was used as the macroinitiator of glycidol polymerization. Four copolymers with the DP of the polyglycidol block equal to 13, 42, 69, and 117 were obtained. Information about PS-*b*-PGL copolymers used in this study are presented in Table 1. A detailed description of their polymerization and parameters was given in our previous work [43].

In Ref. [43], we described the formation of micelles by amphiphilic PS-*b*-PGL copolymers in water. Micellization was performed by dialysis from organic solvents, DMF and dioxane, to water. When copolymers were dialyzed from DMF, which is a good solvent for both blocks, two fractions of particles appeared: typical core-shell micelles and aggregated clusters. In the case of dioxane, a selective solvent for the polystyrene block, one population of particles corresponding to typical regular micelles was observed. Types of PS-*b*-PGL aggregates were therefore dependent on the initial solvent used in dialysis. The average diameter of micelles acquired from dioxane increased with the length of copolymer chain. CMC values were very close for PS-*b*-PGL copolymers in both organic solvents, and it increased linearly with the PGL to PS ratio.

### 3.1. Critical Micelle Concentration of PS-b-PGL in Dioxane

To carry out the micellization of PS-*b*-PGL(X) copolymers, samples were dissolved in dioxane at different concentrations. As for other micellization processes [46,47,48], the behaviour of PS-*b*-PGL(X) copolymers was followed by measuring the scattered light intensity of solution/dispersion as a function of copolymer concentration (Figure 1a).

Due to its limited solubility in dioxane, PS-*b*-PGL(X) block copolymers should form reverse micelles in dioxane with the hydrophilic PGL block in the interior of the aggregated structure. As demonstrated in Figure 1a, the intensity of light scattered was weak for all solutions tested at very low concentrations. Above a certain concentration, the intensity increased and deviated from the straight line dependence. This indicates the beginning of aggregation of the copolymer chains. The CMC was determined as the onset of micellization at the point of intersection of the tangents of plots of intensity as function of copolymer concentration. The values obtained are presented in Table 1.

The block copolymer PS-*b*-PGL(1) with the shortest PGL block did not show a CMC in dioxane in the range of concentrations studied. Increasing the content of hydrophilic glycidol units in the copolymers resulted in decreasing the copolymers’ CMC, as it is indicated in Table 1. This kind of dependence was also found for amphiphilic block copolymers in water. It was reported for polystyrene-*b*-poly(ethylene oxide) [17,49,50] and for triblock copolymers polyglycidol-*b*-poly(propylene oxide)-*b*-polyglycidol [51]. A similar correlation was observed for polystyrene-*b*-poly(ethylene oxide) in ionic liquids [52].

The formation of micellar structures was evidenced by DLS. Figure 1b shows the distribution of diameters of the copolymers for 1 mg/mL. Copolymer PS-*b*-PGL(1) does not form micelles or aggregates in the tested concentration range (from 0.01 to 2 mg/mL). In case of copolymers with longer PGL blocks, only one population of particles can be found in the polymer dispersion. The dispersity of particle diameters (PDI) and their hydrodynamic diameter values (D_h_) are given in Table 2. The diameters of particles were below 20 nm and decreased with increasing the length of the PGL block. Symmetrical and narrow size distributions of the copolymer self-assemblies and their diameters of only a dozen nanometers prove the formation of reversible core-shell PS-*b*-PGL micelles in dioxane.

Table 2 contains the diameters of copolymer particles. The lengths of copolymer chains and PGL blocks, calculated using bound length and angles of copolymer units and assuming planar zig-zag conformation of the chains [51], were added to the table. It can be noticed that hydrodynamic diameters of the micelles decrease with increasing the length of PS-*b*-PGL chains. In all cases, the diameters are less than twice the calculated contour length of the copolymer chains, which confirms the formation of core-shell micelles. The packaging efficiency of PGL in micelle cores increases with the length of PGL block. The compressing of PGL chains and the creation of a surrounding PS shell ensure the colloidal stability of PS-*b*-PGL micelles in dioxane. Because the length of PS block is the same for all PS-*b*-PGL copolymers, the smaller diameters of the micelles with longer PGL blocks are possible only when longer PGL blocks are better packed.

As shown in our previous studies [43], the same copolymers formed the regular micelles in water when dialyzed from dioxane solutions. The PS-*b*-PGL micelles had in their interior PS blocks of the same length. The micelles’ diameter increased with increasing the length of the hydrophilic block to 24, 31, 42 and 56 nm. As can be seen in Table 2, the reverse micelles of PS-*b*-PGL are much smaller, and the dependence of the size of micelles and polymer chain lengths is opposite to that for regular PS-*b*-PGL micelles in water.

Visualization of PS-*b*-PGL micelles was done using the TEM technique. Microphotographs (Figure 2) revealed nearly spherical particles with diameters corresponding to those obtained from DLS measurements. In TEM, the diameters of the aggregates were about 22 nm for PS-*b*-PGL(2), 19 nm for PS-*b*-PGL(3), and 17 nm for PS-*b*-PGL(4). All diameters given are an average of values taken for about 50 species.

### 3.2. Aggregation of PS-b-PGL Copolymers in Water/Dioxane Mixtures

Phase separation of polystyrene and polyglycidol blocks of PS-*b*-PGL was performed by the addition of water to dioxane copolymer solutions. Initial concentrations of copolymers in dioxane were far below their CMC values in the solvent, which assured separation of individual polymer chains. The presence of water in solution leads to a decrease in the solubility of the PS block and an increase in the solubility of the PGL block. When water concentration exceeds the critical water content, the copolymer chains self-assemble.

Figure 3 shows the relationship between the intensity of the scattered light of copolymer solutions as a function of water content in dioxane/water mixtures.

For all investigated copolymers and all initial copolymer concentrations, when the fraction of water added in the solution was low, the intensity of scattered light was also very low (Figure 3). This indicates that the polymer chains are isolated in solution. Addition of water causes a rapid increase in the intensity of the scattered light for all investigated solutions. This indicates that phase separation did occur, and the copolymer began to aggregate. Above a certain value of water content in solution, the intensity of scattered light approaches a plateau. The critical water content was determined from the intersection of the tangents on the plot of the intensity of scattered light as a function of water content.

The relationship between the CWC and the initial concentration of copolymer in solution expressed in a logarithmic scale is shown in Figure 4a. The dependence could be approximated by straight lines for all the tested copolymers. Figure 4b shows plots of the CWCs of copolymer solutions as function of the length of the copolymers’ hydrophilic block. For a chosen initial concentration of the copolymers, CWC increases proportionally to the PGL/PS ratio. The straight lines are parallel for all studied initial concentrations. Similar relationships between CWC vs. initial concentration and CWC vs. copolymer block ratio were reported by Eisenberg et al. for a series of polystyrene-*b*-poly(acrylic acid) copolymers [5,9,10,11]. The decrease of CWC with increasing concentration of a polymer in the initial solution could be due to the lower amount of water necessary to cross the threshold at which aggregation begins to occur.

Diameters of aggregates were measured by DLS when they were stable above the CWC. The values for aggregates formed in solution with 50% water content were used for the analysis discussed below.

Figure 5 shows the size distributions of copolymer aggregates formed in 50/50 dioxane/water solutions. In all cases, the particles become smaller with higher initial copolymer concentrations. The particle size distribution is the narrowest for the particles formed at the highest initial copolymer concentration. In the distributions for lower concentrations, a tail extending to higher sizes could be observed. Such tailing could be due to the presence of a particle population with non-spherical morphology. Different morphology of aggregates was reported in the studies of poly(4-vinyl pyridine)-*b*-polystyrene-*b*-poly(4-vinyl pyridine) in dioxane/water mixtures, depending on water content [53].

The D_h_ of particles exceeded 300 nm for PS-*b*-PGL, with 13 DP of polyglycidol block in case of the lowest initial concentration. For the highest studied initial copolymer concentrations, particles smaller than 50 nm were created from all investigated copolymers. Taking into account the contour length of copolymer chains (Table 2), it is highly probable that at this initial concentration, the copolymers with polyglycidol blocks longer than 13 units form regular core-shell micelles.

The dependence of the average diameters of particles on the initial PS-*b*-PGL concentration in dioxane is shown in Figure 6a. Whereas, the similar dependence of particle diameters on the PGL/PS ratio is shown in Figure 6b.

The plots illustrating the decrease of diameters of particles as function of the initial copolymer concentration (expressed in logarithmic scale) could be well fitted by straight lines. A similar dependence was observed for block copolymers of polystyrene and poly(acrylic acid) [11,54,55]. The authors associated this with formation of a different morphology by the copolymers studied.

The degree of polymerization of the hydrophilic polyglycidol block has a clear impact on the diameter of aggregates (Figure 6b). With the increasing initial concentration of the copolymers, the particle diameters decreased linearly with increasing the length of the polyglycidol block. For the highest content of glycidol units in copolymer, the aggregate sizes were the smallest. In the case of the highest studied initial concentration, aggregate diameters decreased slightly with the PGL/PS ratio ranging from 39 nm for PGL/PS = 0.32 to 30 nm for PGL/PS = 3.43.

## 4. Conclusions

This work reports the results of studies on the behaviour of amphiphilic PS-*b*-PGL copolymers, which differ in the length of the polyglycidol block, in dioxane and dioxane/water mixtures. The general conclusion is that for copolymers with constant DP of PS the micellization of PS-*b*-PGL strongly depends on DP of the PGL blocks and on water content in dioxane/water mixtures as a result of PGL interaction with water, presumably due to hydrogen bonding. For the copolymer with the shortest PGL block (DP = 13) and a PS block with DP equal to 29, micellization of its chains in dioxane was not observed for copolymer concentrations up to 2 g/mL. Three other copolymers formed reverse core-shell micelles in dioxane. Their CMC decreased with increasing the length of the PGL block in the copolymer chain. Micelle diameter also decreased, indicating increasing efficiency of PGL compression in micelle cores.

The particle formation from PS-*b*-PGL copolymers was performed by addition of water to the copolymer solution in dioxane (CWC process). Critical water content decreased with increasing the initial concentration of the copolymers. However, it increased as the length of the hydrophilic PGL block increased. Particles resulting from the addition of water to dioxane became smaller when the initial copolymer concentration was increased. For the highest concentration used in the studies, the size of the aggregated structures of the copolymers in water/dioxane solution were below 40 nm, which is close to the sizes of the copolymer micelles formed in water [43]. The diameters of copolymer particles were linearly dependent on the PGL/PS ratio in the copolymer chains. The more hydrophilic copolymers formed smaller particles.

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
