# Peer review of "Micellization of Polystyrene-b-Polyglycidol in Dioxane and Water/Dioxane Solutions"

_polymers, 2020, doi:10.3390/polym12010200_

Round 1
Reviewer 1 Report
In this manuscript, the author reported the self-assembly of a series of amphiphilic polystyrene-b-polyglycidol (PS-b-PGL) diblock copolymers in dioxane and dioxane/water mixtures. The micelle size, CMC, and CWC were studied. The following concerns should be addressed before further evaluation.
Most importantly, the reverse spherical micelles of PS-b-PGL in toluene has been studied by other groups. In this work, the authors studied the behavior of PS-b-PGL in dioxane and dioxane/water. What are the novelty and new findings? Why not other organic solvents? The polymers were synthesized by anionic polymerization. They are polydispersed. It is not right to use such words as “fixed length” “precisely controlled number” and it should use word such as average length. The authors should keep to use abbreviations after the initial define. PS-b-PGL and polystyrene-b-polyglycidol are both used in the current form. The authors should carefully revise all the figures. Some of the curve and labeling is not visible.
Reviewer 2 Report
The work submitted by Trzebicka and coworkers summarises the main results on the self-assembly of polystyrene-b-polyglycidol copolymers in dioxane and in dioxane/water solvent mixtures. This manuscript closely follows earlier work of these authors (ref. 43, Eur. Polym. J. 99 (2018) 72-79).
The manuscript is in general well written and organised. The topic is of soundness and fits to the scope of this journal. I think these results are of good interest, and can be published after minor revisions. I had only a minor comment regarding the presentation and I suggest including a figure presenting the molecular formulae of studied copolymers.
